# Bacterial Colonization and Proliferation in Furcal Perforations Repaired by Different Materials: A Confocal Laser Scanning Microscopy Study

**Shlomo Elbahary** [1,*,†], **Sohad Haj Yahya** [1,†], **Cemre Koç** [2], **Hagay Shemesh** [3], **Eyal Rosen** [1,†] and **Igor Tsesis** [1,†]

[1] Department of Endodontology, Tel Aviv University Dental School, Tel Aviv 6997801, Israel;
s_sohad@yahoo.com (S.H.Y.); dr.eyalrosen@gmail.com (E.R.); zasis@post.tau.ac.il (I.T.)

[2] Department of Endodontics, Faculty of Dentistry, Başkent University, Ankara 06490, Turkey;
cemrekoc@baskent.edu.tr

[3] Department of Endodontology, Academic Centre of Dentistry Amsterdam (ACTA),
University of Amsterdam and VU University, 1008 Amsterdam, The Netherlands; H.Shemesh@acta.nl

\* Correspondence: shlomoll@tauex.tau.ac.il; Tel.: +972-548300393

† These authors contributed equally to this work.

**Abstract:** Following furcal perforation, bacteria may colonize the defect and cause inflammation and periodontal destruction. This study used confocal laser scanning microscopy (CLSM) to evaluate *Enterococcus faecalis* colonization and proliferation in furcal perforations repaired with different materials. Furcal perforations created in 55 extracted human mandibular molars were repaired using either MTA-Angelus, Endocem, or Biodentine and coronally subjected to *E. faecalis* suspension for 21 days. The specimens were then stained using a LIVE/DEAD Viability Kit and visualized by CLSM. The minimum and maximum depths of bacterial penetration into the dentinal tubules were 159 and 1790 μM, respectively, with a mean of 713 μM. There were significantly more live than dead bacteria inside the dentinal tubules ($p = 0.0023$) in all groups, and all three repair materials exhibited a similarly sized stained area ($p = 0.083$). However, there were significant differences in the numbers of dead bacteria at the circumference of the perforation defect ($p = 0.0041$), with a significantly higher ratio of live to dead bacteria in the MTA-Angelus group ($p = 0.001$). Following perforation repair, bacteria may colonize the interface between the repair material and dentin and may penetrate through the dentinal tubules. The type of repair material has a significant effect on the viability of the colonizing bacteria.

**Keywords:** bacterial colonization; confocal laser scanning; *Enterococcus faecalis* microscopy; perforation; repair materials

## 1. Introduction

Perforation can be defined as an artificial communication between the root canal space and the surrounding tooth tissue (periodontium), or oral environment [1]. Perforations can be pathological (caused by root resorption or caries) or iatrogenic (as a result of dental procedures during access cavity preparation, canal negotiation, or post space preparation) [1,2]. Various factors, such as time elapsed before perforation repair, location or size of the perforation, the repair material used, and the experience of the operator, may all affect the treatment outcome [1,3–5].

The main goal of perforation management is to seal the defect in order to prevent bacterial contamination, inflammation, and loss of periodontal attachment and to prepare an optimal environment for tissue repair. In this context, the sealing ability and marginal adaptation of the repair material used are crucial in preventing the leakage of irritants and thus enhancing the chances of success [6–8]. A wide variety of materials including MTA, amalgam, glass ionomer cement, intermediate restorative material, and tricalcium phosphate have been tried in furcal and root perforation repair [4,7–11]. Notably, the

use of biocompatible materials in perforation repair has been associated with a lower inflammatory response in the surrounding tissues [3,9].

The ability of different restorative materials to repair the perforation defect has been assessed by various in vitro experimental methods and reagents, including the bacterial leakage model [12], radioisotopes [13], dye penetration [11], and a fluid filtration method [14]. However, the reliability of these techniques has been questioned due to the fact that leakage may occur not only through the interface between the material and the dentine wall but also through other possible areas. This may be overlooked if the appropriate negative controls are not included. In addition, these methods do not provide information about the extent of bacterial penetration into the dentin. Significantly, while histological sections can be used to identify the presence and distribution of bacteria in the dentin tubules, they fail to evaluate the viability of the bacteria.

These failings mean that alternative microscopic techniques are necessary in order to analyze more precisely the bacterial leakage and penetration depth into the dentinal tubules. Confocal laser scanning microscopy (CLSM) has been considered as an alternative microscopic technique that can provide quantitative and clinically relevant data about the presence or absence of bacteria and the extent to which the bacteria have colonized the sides of the dentinal tubules and root canal walls [15–17]. Thus, CLSM has an advantage over conventional in vitro experimental settings in comparing the bacterial colonization of perforation defects that have been repaired with various materials [18].

In this study, we used CLSM to assess the colonization and proliferation of *Enterococcus faecalis* in furcal perforations, repaired with MTA-Angelus, Endocem, or Biodentine. The null hypothesis was that bacterial penetration does not depend on the type of repair material.

## 2. Materials and Methods

### 2.1. Preparation of Specimens

The study was approved by the Ethics Committee of Tel-Aviv University (No: 230.17, 23 January 2018), and all protocols were conducted in accordance with the relevant regulations and guidelines.

Fifty-five freshly extracted human permanent molars (for periodontal reasons) were used in this study. Teeth with previous endodontic treatment, visible sign of root resorption, caries, root fractures, or immature apices, were excluded from the study. Teeth with fused roots or with a single root were also excluded. Once cleaned of debris and soft tissue remnants, teeth were stored in phosphate-buffered saline.

For use, the teeth were decoronated to expose the pulp chamber cavities, and 5 mm of the apical segment of the roots was amputated using a high-speed diamond disc (IPR Diamond Disc; Dentsply Int./Maillefer, Ballaigues, Switzerland) with water cooling. Pulpal remnants were removed from pulp chamber and root canals using a barbed broache (Barbed Broache; Dentsply Int./Maillefer, Ballaigues, Switzerland). The canal orifices and apical foramen of each root were filled with resin composite (TE; Ivoclar Vivadent AG, Schaan, Liechtenstein, German). Perforations were created in the center of the pulpal floor by using a round #14 diamond bur while the tooth was visualized under an optical microscope (OPMI pico Dental Surgical Microscope, Carl Zeiss Meditec, Dublin, CA, USA) at 6 × magnification [19].

The specimens were randomly divided into the following experimental groups:

(a) Group 1 (*n* = 10): The perforation defects were repaired using MTA-Angelus (Angelus, Londrina, PR, Brazil), mixed according to the manufacturer's recommendations. A metal spatula was used to mix a 1:1 ratio of MTA Angelus with distilled water on a sterilized glass slab. The mixture was homogeneous and had a consistency similar to wet sand.

(b) Group 2 (*n* = 10): The perforation defects were repaired using Endocem MTA (Maruchi, Wonju, Korea) cement, mixed according to the manufacturer's recom-

mendations. A metal spatula was used to mix 300 mg Endocem MTA powder with 0.12 mL liquid on a sterilized glass slab.

(c) Group 3 (*n* = 10): The perforation defects were repaired using Biodentine (Septodont, Saint-Maur-des-Fossés, France), mixed according to the manufacturer's recommendations. The closed capsule was gently tapped on a hard surface to dispense the powder. Five drops of liquid were added into the capsule and mixed in a triturator for 30 s.

(d) Group 4 (*n* = 5) (positive control): The created perforations were left without repair material.

(e) Group 5 (*n* = 5) (negative control): The teeth were left without perforation and repair material, but the external tooth surface was covered with two layers of nail varnish.

(f) Group 6 (*n* = 5) (negative control MTA Angelus): The teeth were the same as group 1 without bacterial contamination.

(g) Group 7 (*n* = 5) (negative control Endocem MTA): The teeth were the same as group 2 without bacterial contamination.

(h) Group 8 (*n* = 5) (negative control Biodentine): The teeth were the same as group 3 without bacterial contamination.

All teeth were left at 37 °C and 100% humidity for 24 h. All specimens were prepared by a single operator.

### 2.2. The Experimental Setting

To avoid any bacterial leakage through accessory canals or other discontinuities in the cementum, all external surfaces of the teeth, apart from the perforation site, were coated with two layers of nail varnish. According to a model described previously, all specimens were placed in the glass vials (Sigma-Aldrich, St. Louis, MO, USA) through the rubber cap. Cyanoacrylate adhesive was used to seal the interface between the specimen and the rubber cap (Krazy Glue; Krazy Glue, Columbus, OH, USA) (Figure 1a) [20,21].

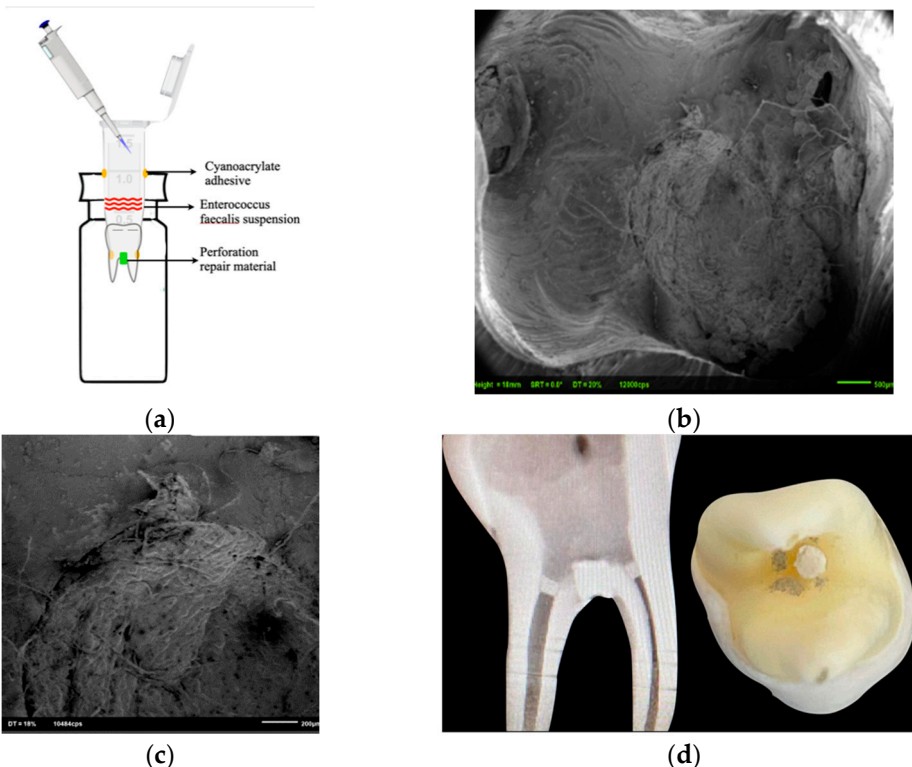

(a)

(b)

(c)

(d)

**Figure 1.** (**a**): Illustration of the experimental model. (**b**,**c**): SEM images displaying bacterial penetration at the perforation site. Clinical and X-ray of lower molar after perforation repair (**d**).

### 2.3. Simulation of E. faecalis Contamination

The specimens were sterilized overnight with ethylene oxide gas. *Enterococcus faecalis* (ATCC® 29212™) growth medium was prepared and then autoclaved. Media were supplemented with 0.5 mg/mL streptomycin sulfate (Streptomycin sulfate; Sigma-Aldrich, St. Louis, MO, USA) to prevent contamination by additional bacterial species. A freshly prepared bacterial suspension was added to the coronal side of the specimen and incubated at 37 °C and 100% humidity. The bacterial suspension was replaced every 24 h for a total of 21 days.

### 2.4. Confocal Laser Scanning Microscopy Analysis

After the incubation period, the specimens were embedded in a self-cure acrylic (Unifast Trad, Alsip, IL, USA). They were cut perpendicularly to the long axis of the root, through the perforation site containing the repair materials, using a diamond saw at 500 rpm (Isomet; Buehler, Lake Bluff, IL, USA) under continuous water irrigation. Viability was assessed by staining the specimens with the LIVE/DEAD BacLight Bacterial Viability kit L-7012 (LIVE/DEAD BacLight Bacterial Viability Kit for microscopy and quantitative assays; Molecular Probes, Eugene, OR, USA), including separate vials of the two component dyes (1:1 mixture of SYTO 9 and propidium iodide). The excitation/emission maxima for these dyes are 480–500 nm for the SYTO 9 stain (live bacteria stained in green) and 490–635 nm for propidium iodide (dead bacteria stained in red) [22]. Environmental SEM (ESEM) was acquired in the environmental "wet" mode by using a Philips XL30 ESEM-Feg (FEI/Philpips Electron Otpics, Eindhoven, The Netherlands, operating conditions: 5 °C, 2.9–5.9 torr gas pressure, 80% relative humidity, 6–9 kV) to scan one slice from each tooth in order to validate the bacterial leakage model. Five interesting spots on each specimen were chosen (Figure 1b,c).

Fluorescence from the stained areas was observed immediately under a CLSM (Leica TCS SP5; Leica Microsystems CMS, Mannheim, Germany). Simultaneous and single channel imaging were used to record the green and red fluorescence.

The CLSM images of materials and dentinal tubules were recorded at a resolution of $1024 \times 1024$ pixels. All the extensions of contaminated dentinal tubules of the specimens were analyzed using the software (LAS AF software, version 2.6.0.7266; Leica Microsystems CMS, Germany) [18,23,24]. First, the size of fluorescent staining area in the related region was measured, and then the penetration of bacteria into the dentinal tubules was evaluated by considering the canal wall as the starting point. The ratio of the green and red stained areas represented the viability of the bacteria, and this parameter was used to evaluate the effect of the repair material employed.

### 2.5. Statistical Analysis

Mead's resource equation was used in order to estimate the sample size. One-way ANOVA was performed in order to assess the size of fluorescent stained areas and the penetration depth of bacteria into the dentinal tubules with the various perforation repair materials. Pearson's chi-squared test was also used to analyze dependency between bacteria being alive or dead and the material used for perforation repair. The significance level was set at $\alpha = 0.05$. The SPSS program (IBM Corp., 2011, Version 21.0. Armonk, NY, USA) was used for the statistical analysis.

## 3. Results

No fluorescence was detected in the negative control groups (Groups 5–8), but the positive control group (Group 4) did indeed exhibit positive fluorescence. There were no significant differences ($p = 0.15$) between the stained areas in the buccal, lingual, mesial, and distal directions for the experimental groups (Groups 1–3, Figure 2). Similarly, there were no significant differences between the repair materials with respect to the stained regions in the buccal, lingual, mesial, and distal areas (in total cells or in the live to dead ratio, $p = 0.083$).

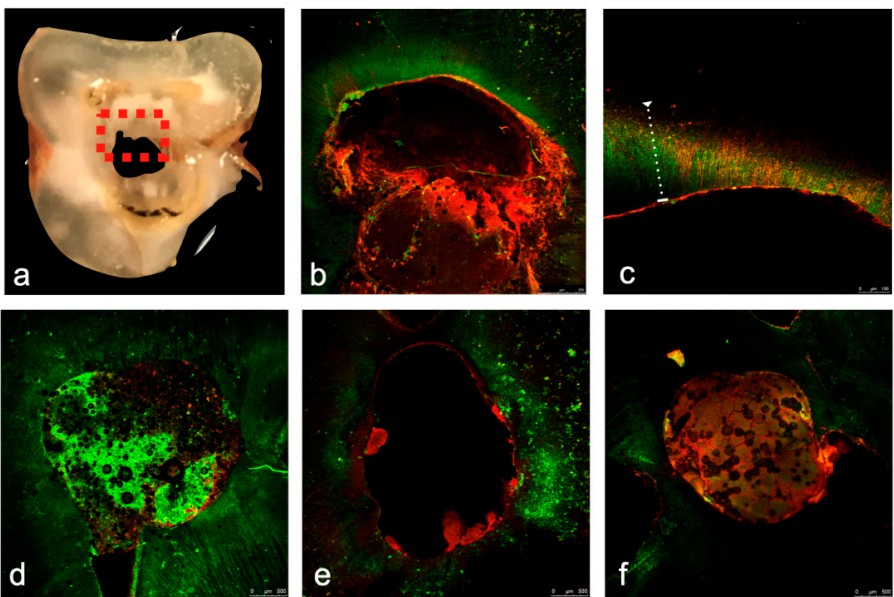

**Figure 2.** (**a**–**f**). Confocal laser scanning microscopy (CLSM) images. The specimens were cut through the perforation site containing the repair materials perpendicularly to the long axis of the root in order to be evaluated by the CLSM. The perforation site can be clearly seen, and the dotted square represents a single evaluated area (**a**). Prior to scanning, LIVE/DEAD Kit stained the infected dentin. Vital (**Green**) and dead (**Red**) bacteria inside the dentinal tubules in the evaluated areas are clearly visible (**b**–**f**). Positive control (**b**,**c**), MTA-Angelus (**d**), Endocem (**e**) and Biodentine (**f**) groups showing fluorescence staining. Significantly more dead bacteria (**Red Staining**) than live bacteria (**Green Bacteria**) were detected on the dentinal surface at the circumference of the perforation defect (**d**–**f**), and significantly more live (**Green Staining**) bacteria than dead bacteria down in the dentin inside the dentinal tubules (**b**–**f**) were detected in all groups.

There were significantly more dead bacteria than live bacteria on the dentinal surface at the circumference of the perforation defect ($p = 0.0041$) and significantly more live bacteria than dead bacteria down in the dentin inside the dentinal tubules, for all directions ($p = 0.0023$) in all groups. (Figure 3). The ratio of live to dead bacteria was significantly higher in the MTA-Angelus group when compared with the other repair material groups ($p = 0.001$).

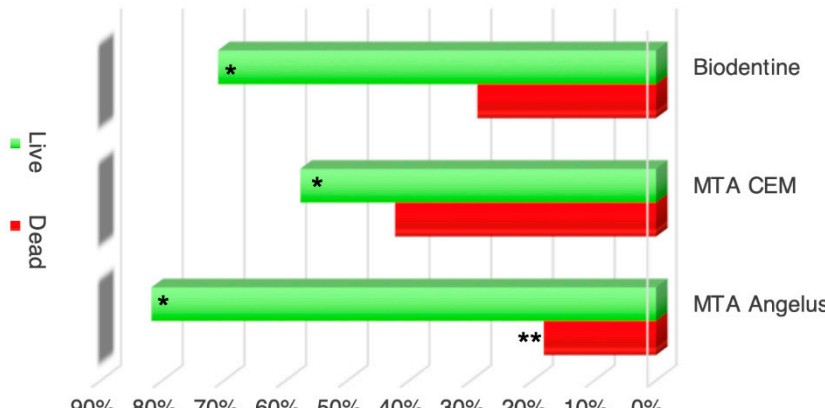

**Figure 3.** The average percentages of live (**Green**) and dead bacteria (**Red**). The average bacterial contamination (stained areas) of live and dead bacteria in all locations tested combined (buccal, lingual, mesial, and distal). * Significantly more live bacteria were seen in Groups 1–3. ** The ratio of live to dead bacteria was significantly higher in the MTA-Angelus group (Group 1) than in the other groups ($p = 0.001$).

No significant differences were observed between the tested repair materials in terms of extent of bacterial penetration through the dentinal tubules ($p = 0.277$). The minimum and maximum penetration depths were 159 and 1790 µM, respectively, with a mean of 713 µM (Table 1).

**Table 1.** Bacterial Penetration depth (in um) in the different groups.

| | Quartile | | | Min | Max | Median | Stdv | *p*-Value |
|---|---|---|---|---|---|---|---|---|
| | Q1 | Q2 | Q3 | | | | | |
| MTA-Angelus | 862 | 987 | 1295 | 159 | 1790 | 987 | 455 | $p = 0.092$ |
| MTA CEM | 325 | 432 | 456 | 211 | 782 | 832 | 176 | $p = 0.229$ |
| Biodentine | 467 | 720 | 930 | 224 | 1641 | 720 | 289 | $p = 0.079$ |

## 4. Discussion

Understanding the sealing ability of materials used in the repair of perforations and predicting the amount and direction of bacterial leakage are of paramount importance in addressing the pathologies related to perforations [25,26]. CLSM could be considered a useful modality of choice, in addition to traditional standard electron microscopy and PCR-based techniques, to identify viable bacteria in dentinal tubules [16,17,24,27]. Using CLSM together with the live/dead staining method allows us to assess the extent of contamination and the viability of bacteria in contaminated dentinal tubules [16,17,24]. To our knowledge, this is the first study to use CLSM to assess bacterial colonization in perforation sites of extracted human molar teeth repaired with three different materials. In contrast to previous reports using bacterial leakage models, our current study provides histological evidence for the actual routes of bacterial colonization. One positive and four negative histological controls were used to confirm the validity of the experimental model.

After perforation occurs, the success of the treatment relies on suitably placed repair material designed to prevent communication between the root canal space, peri-radicular tissues, and oral environment [28–32]. MTA is a hydrophilic calcium silicate-based material and is compatible with moist conditions such as those found in perforation sites [33]. Additional favorable properties include biocompatibility, non-cytotoxicity, radiopacity, availability, and promotion of tissue regeneration [8,34–37], as well as induction of cementogenesis and osteogenesis [11,12]. However, the material also has a number of disadvantages, such as the long setting time, difficulties in manipulation, and potential for tooth discoloration [38–40]. Recently, novel calcium silicate-based materials have been introduced in order to overcome these MTA shortcomings [41,42].

Biodentine is a widely used bioactive material consisting of tricalcium silicate, calcium carbonate, zirconium oxide, and calcium chloride. Biodentine has been reported not only to have improved sealing ability and a short setting time [42,43] but also to possess bioactivity and biomineralization properties [44–48].

Endocem MTA is another recently introduced fast-setting calcium silicate-based cement and is composed of calcium oxide, silicate oxide, aluminum oxide, and bismuth trioxide. Its fast-setting property depends on the presence of fine particles of pozzolan, which is a siliceous and/or aluminous material. Endocem MTA exhibits acceptable biocompatibility, induces reasonable mineralization, and has less discoloration potential than traditional MTA [41].

Most bacteria cannot survive at an alkaline pH. Accordingly, the growth of *Enterococcus faecalis* can be suppressed at pH 10.5–11.0, and there was no survival reported above pH 11.5 [47]. Calcium silicate-based materials form a silicate gel at their surface when mixed with water, and any calcium hydroxide in the silicate gel releases hydroxyl ions into the environment, which increases the pH [48,49]. The antibacterial effects of calcium silicate-based materials have generally been attributed to the resulting high pH levels, although this assumption is not totally approved in clinical situations. In the present study, higher numbers of dead bacteria than live bacteria were detected on the

inner dentinal surface of the perforation defect. However, there were more live bacteria than dead bacteria in the dentin inside the dentinal tubules. This can be attributed to the buffering capacity of dentine, which decreases the pH in the dentine tubules as the distance from the perforation defect increases.

To our knowledge, this is the first study to compare the colonization depth of *Enterococcus faecalis* when using MTA-Angelus, Biodentine, and Endocem as perforation repair materials. All the tested materials in the present study were calcium silicate-based. However, unlike MTA-Angelus and Biodentine, Endocem contains fluoride and stannous fluoride, which can inhibit the growth of *Enterococcus faecalis* [50]. This might explain the observation that although no significant differences were found between the tested materials in terms of stained areas, there was a trend to lower values with Endocem repair. A previous study by Tsesis et al. (2018) that compared the bacterial colonization depth of different materials for retrograde filling accords with our results in that MTA-Angelus and Biodentine displayed a similar performance [18]. However, the results of the present study indicate that the type of repair material did have an effect on the viability of the colonizing bacteria, with significantly more live bacteria detected in the MTA-Angelus group than in the Endocem and Biodentine groups. This is in contrast to a previous study by Jardine et al. (2019) who used confocal laser microscopy to evaluate the viability of a multispecies microcosm in the vicinity of bioceramic cements and concluded that MTA-Angelus was as effective as Biodentine in terms of antimicrobial activity [51]. This discrepancy may be explained by the different microorganisms involved. A limitation of the current study is that the use of an in vitro model may not completely recapitulate the exact clinical situation. Several recent studies from 2019–2021 [52–54] attempted to compare the sealing ability of different bioceramic materials, but these used dye extraction [52,53] or protein leakage [54] models without any added bacterial contamination. Further in vivo studies will be needed to assess the antibacterial activity of various calcium silicate-based materials in more detail.

## 5. Conclusions

Taking the limitations of an ex vivo setting into account, the present study demonstrates that bacteria may colonize the interface between the repair material and the dentin walls and may penetrate into the dentinal tubules. Clinicians should choose repair material with care since it has implications for the viability of the colonizing bacterial.

**Author Contributions:** Conceptualization, S.E. and C.K.; methodology, I.T. and E.R.; software, S.H.Y. and S.E.; validation, H.S. and I.T.; formal analysis, I.T.; investigation, S.H.Y. and S.E.; resources, C.K.; data curation, S.H.Y.; writing—original draft preparation, S.E. and S.H.Y.; writing—review and editing, I.T., S.E., and S.H.Y.; visualization, S.E.; supervision, H.S and E.R. All authors have read and agreed to the published version of the manuscript.

**Funding:** This study was supported by the Ernst & Tova Turnheim Clinical Research Fund in Dentistry.

**Institutional Review Board Statement:** The study was approved by the Ethics Committee of Tel-Aviv University (No: 230.17, 23 January 2018), and all protocols were conducted in accordance with the relevant regulations and guidelines.

**Informed Consent Statement:** Informed consent was obtained from all subjects involved in the study.

**Conflicts of Interest:** The authors declare no conflict of interest.

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
