# Peer review of "Bacterial Colonization and Proliferation in Furcal Perforations Repaired by Different Materials: A Confocal Laser Scanning Microscopy Study"

_applsci, doi:10.3390/app11083403_

Round 1

Reviewer 1 Report

The manuscript titled "Bacterial Colonization and Proliferation in Furcal Perforation Repaired by Different Materials: a Confocal Laser Scanning Microscopy Study" focuses on the effect of perforation/materials on the bacterial colonisation using CLSM. The study is short and concise. The hypothesis is justified by the results. The manuscript is well written. I do have very minor doubts.

  1. Was the root canal treatment and filling done by one or more operator?
  2. Kindly add pictures of the root canal filled teeth(Xray).

Reviewer 2 Report

This manuscript shows an in vitro study to assess colonization and proliferation of Enterococcus Faecalis in molars furcal perforations repaired with three different materials commercially available. To do that, extracted teeth were studied by using confocal laser scanning microscopy. Despite I think that this is an interesting study with an appropriate design, I have some concerns that authors should addressed. In my opinion, the results are poorly presented and authors must make a big effort to improve this part of the manuscript. In addition, description of the methods can be improved. Furthermore, after rewriting the results and modifying the discussion accordingly. Besides, the whole manuscript must be proofread by an English native speaker specialized in scientific text. English is not my native language either, but I have found many errors that any current scientific proofreading company can correct before any resubmission.

My main concerns about this manuscript are:

  • The methods described 8 different experimental groups. However, authors state "all groups" to refer only to the three materials tested (groups 6-8). This is inappropriate.
  • Figure 2 include, I guess, representative CLSM images from the positive control group (group 4) and the three materials tested (groups 6-8). However, this is difficult to identify for readers. Authors must include some little information on each picture (e.g. #group to facilitate the understanding of the figure and may indicate by means of arrows or other signs where to locate the described results). For me, images d, e and f are completely different. However, from the caption I seem to understand that the results in the 3 experimental groups are similar. I am sorry but I do not see or understand this. Authors must be improve the presentation of their result.
  • Figure captions should explain properly its content. Figure 3 apparently only shows the average percentages of live and dead bacteria and authors do not specify any location. However, authors state in the results section that such percentages varies depending on the location: “the dentinal surface at the circumference of the perforation defect” and “the depth of the dentin inside the dentinal tubules for all directions”. So, which location is referring the Figure 3? The authors should show the graphics from all data. Furthermore, the meaning of the asterisk (*) or the double asterisk (**) should be defined in figure captions.

Reviewer 3 Report

The topic of the manuscript is the evaluation of E. faecalis colonization and proliferation in furcal perforations repaired with different materials (such as MTA Angelus, Endocem and Biodentine), by using confocal laser scanning microscopy.

The abstract and the main text of the article are informative. The Introduction clearly presents the clinical problem of perforations. The section “Material and Methods” very precisely explains the chosen study design. The sections “Results” and “Discussion” are interestingly written, however can be improved.

Some following points must be clarified/corrected for the further processing of this article.

Merits-related comments:

  1. “The significance level was set at p<0.05.” should be replaced by “The significance level was set at α=0.05.”.
  2. In the case of continuous variable, such as bacterial penetration depth, compliance with the normal distribution must be checked, e.g. by Shapiro-Wilk test. The compared variables probably do not have a normal distribution, and then non-parametric test should be used instead of parametric one. Furthermore, they should be presented as medians and quartiles (not means and standard deviations) - the median and standard deviation must not be combined in the results, as was done in Table 1.
  3. It is suggested to add more recent articles from 2016-2021 to the references in the Discussion.
  4. The conclusions should be more concrete “take-home” messages.

Technical comments:

  1. The abstract should be a single paragraph and follow the style of structured abstracts but without headings.
  2. The main text should not be bolded in random places.
  3. “Statistical analysis” should be numbered as 2.5, as well as “Results”, “Discussion” and “Conclusions” respectively 3, 4 and 5.
  4. Meticulous correction of typos recommended.
  5. The citation list must be corrected. References should be described as follows:
    1. Author 1, A.B.; Author 2, C.D. Title of the article. Abbreviated Journal Name YearVolume, page range.
  6. In Author Contributions the following statements should be used "Conceptualization, X.X. and Y.Y.; Methodology, X.X.; Software, X.X.; Validation, X.X., Y.Y. and Z.Z.; Formal Analysis, X.X.; Investigation, X.X.; Resources, X.X.; Data Curation, X.X.; Writing – Original Draft Preparation, X.X.; Writing – Review & Editing, X.X.; Visualization, X.X.; Supervision, X.X.; Project Administration, X.X.; Funding Acquisition, Y.Y.”, and all co-authors must be included.

Round 2

Reviewer 2 Report

The manuscript has improved substantially. Now the text is more concise and readable. I accept it as it is.